# Multimodal Residual Learning for Visual QA

**Jin-Hwa Kim**    **Sang-Woo Lee**    **Donghyun Kwak**    **Min-Oh Heo**
Seoul National University
{jhkim,slee,dhkwak,moheo}@bi.snu.ac.kr

**Jeonghee Kim**        **Jung-Woo Ha**
Naver Labs, Naver Corp.
{jeonghee.kim,jungwoo.ha}@navercorp.com

**Byoung-Tak Zhang**
Seoul National University & Surromind Robotics
btzhang@bi.snu.ac.kr

## Abstract

Deep neural networks continue to advance the state-of-the-art of image recognition tasks with various methods. However, applications of these methods to multimodality remain limited. We present Multimodal Residual Networks (MRN) for the multimodal residual learning of visual question-answering, which extends the idea of the deep residual learning. Unlike the deep residual learning, MRN effectively learns the joint representation from vision and language information. The main idea is to use element-wise multiplication for the joint residual mappings exploiting the residual learning of the attentional models in recent studies. Various alternative models introduced by multimodality are explored based on our study. We achieve the state-of-the-art results on the Visual QA dataset for both *Open-Ended* and *Multiple-Choice* tasks. Moreover, we introduce a novel method to visualize the attention effect of the joint representations for each learning block using back-propagation algorithm, even though the visual features are collapsed without spatial information.

## 1   Introduction

Visual question-answering tasks provide a testbed to cultivate the synergistic proposals which handle multidisciplinary problems of vision, language and integrated reasoning. So, the visual question-answering tasks let the studies in artificial intelligence go beyond narrow tasks. Furthermore, it may help to solve the real world problems which need the integrated reasoning of vision and language.

Deep residual learning [6] not only advances the studies in object recognition problems, but also gives a general framework for deep neural networks. The existing non-linear layers of neural networks serve to fit another mapping of $\mathcal{F}(\mathbf{x})$, which is the residual of identity mapping $\mathbf{x}$. So, with the shortcut connection of identity mapping $\mathbf{x}$, the whole module of layers fit $\mathcal{F}(\mathbf{x}) + \mathbf{x}$ for the desired underlying mapping $\mathcal{H}(\mathbf{x})$. In other words, the only residual mapping $\mathcal{F}(\mathbf{x})$, defined by $\mathcal{H}(\mathbf{x}) - \mathbf{x}$, is learned with non-linear layers. In this way, very deep neural networks effectively learn representations in an efficient manner.

Many attentional models utilize the residual learning to deal with various tasks, including textual reasoning [25, 21] and visual question-answering [29]. They use an attentional mechanism to handle two different information sources, a query and the context of the query (*e.g.* contextual sentences

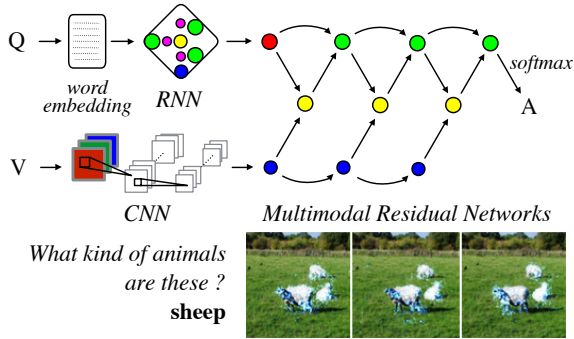

Figure 1: Inference flow of Multimodal Residual Networks (MRN). Using our visualization method, the attention effects are shown as a sequence of three images. More examples are shown in Figure 4.

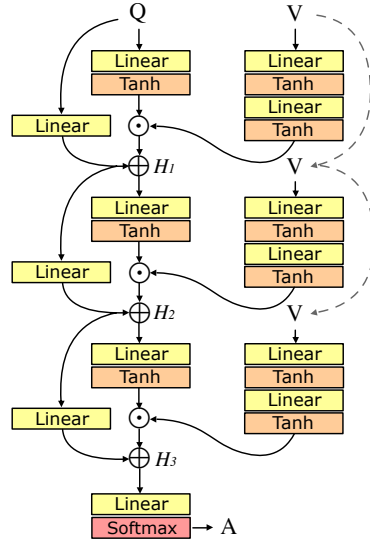

Figure 2: A schematic diagram of Multimodal Residual Networks with three-block layers.

or an image). The query is added to the output of the attentional module, that makes the attentional module learn the residual of query mapping as in deep residual learning.

In this paper, we propose Multimodal Residual Networks (MRN) to learn multimodality of visual question-answering tasks exploiting the excellence of deep residual learning [6]. MRN inherently uses shortcuts and residual mappings for multimodality. We explore various models upon the choice of the shortcuts for each modality, and the joint residual mappings based on element-wise multiplication, which effectively learn the multimodal representations not using explicit attention parameters. Figure 1 shows inference flow of the proposed MRN.

Additionally, we propose a novel method to visualize the attention effects of each joint residual mapping. The visualization method uses back-propagation algorithm [22] for the difference between the visual input and the output of the joint residual mapping. The difference is back-propagated up to an input image. Since we use the pretrained visual features, the pretrained CNN is augmented for visualization. Based on this, we argue that MRN is an implicit attention model without explicit attention parameters.

Our contribution is three-fold: 1) extending the deep residual learning for visual question-answering tasks. This method utilizes multimodal inputs, and allows a deeper network structure, 2) achieving the state-of-the-art results on the Visual QA dataset for both *Open-Ended* and *Multiple-Choice* tasks, and finally, 3) introducing a novel method to visualize spatial attention effect of joint residual mappings from the collapsed visual feature using back-propagation.

## 2 Related Works

### 2.1 Deep Residual Learning

Deep residual learning [6] allows neural networks to have a deeper structure of over-100 layers. The very deep neural networks are usually hard to be optimized even though the well-known activation functions and regularization techniques are applied [17, 7, 9]. This method consistently shows state-of-the-art results across multiple visual tasks including image classification, object detection, localization and segmentation.

This idea assumes that a block of deep neural networks forming a non-linear mapping $\mathcal{F}(\mathbf{x})$ may paradoxically fail to fit into an identity mapping. To resolve this, the deep residual learning adds $\mathbf{x}$ to $\mathcal{F}(\mathbf{x})$ as a shortcut connection. With this idea, the non-linear mapping $\mathcal{F}(\mathbf{x})$ can focus on the

residual of the shortcut mapping $\mathbf{x}$. Therefore, a learning block is defined as:

$$\mathbf{y} = \mathcal{F}(\mathbf{x}) + \mathbf{x} \tag{1}$$

where $\mathbf{x}$ and $\mathbf{y}$ are the input and output of the learning block, respectively.

## 2.2 Stacked Attention Networks

Stacked Attention Networks (SAN) [29] explicitly learns the weights of visual feature vectors to select a small portion of visual information for a given question vector. Furthermore, this model stacks the attention networks for multi-step reasoning narrowing down the selection of visual information. For example, if the attention networks are asked to find a pink handbag in a scene, they try to find pink objects first, and then, narrow down to the pink handbag.

For the attention networks, the weights are learned by a question vector and the corresponding visual feature vectors. These weights are used for the linear combination of multiple visual feature vectors indexing spatial information. Through this, SAN successfully selects a portion of visual information. Finally, an addition of the combined visual feature vector and the previous question vector is transferred as a new input question vector to next learning block.

$$\mathbf{q}^k = \mathcal{F}(\mathbf{q}^{k-1}, \mathbf{V}) + \mathbf{q}^{k-1} \tag{2}$$

Here, $\mathbf{q}^l$ is a question vector for $l$-th learning block and $\mathbf{V}$ is a visual feature matrix, whose columns indicate the specific spatial indexes. $\mathcal{F}(\mathbf{q}, \mathbf{V})$ is the attention networks of SAN.

# 3 Multimodal Residual Networks

Deep residual learning emphasizes the importance of identity (or linear) shortcuts to have the non-linear mappings efficiently learn only residuals [6]. In multimodal learning, this idea may not be readily applied. Since the modalities may have correlations, we need to carefully define joint residual functions as the non-linear mappings. Moreover, the shortcuts are undetermined due to its multimodality. Therefore, the characteristics of a given task ought to be considered to determine the model structure.

## 3.1 Background

We infer a residual learning in the attention networks of SAN. Since Equation 18 in [29] shows a question vector transferred directly through successive layers of the attention networks. In the case of SAN, the shortcut mapping is for the question vector, and the non-linear mapping is the attention networks.

In the attention networks, Yang et al. [29] assume that an appropriate choice of weights on visual feature vectors for a given question vector sufficiently captures the joint representation for answering. However, question information weakly contributes to the joint representation only through coefficients $\mathbf{p}$, which may cause a *bottleneck* to learn the joint representation.

$$\mathcal{F}(\mathbf{q}, \mathbf{V}) = \sum_i \mathbf{p}_i \mathbf{V}_i \tag{3}$$

The coefficients $\mathbf{p}$ are the output of a nonlinear function of a question vector $\mathbf{q}$ and a visual feature matrix $\mathbf{V}$ (see Equation 15-16 in Yang et al. [29]). The $\mathbf{V}_i$ is a visual feature vector of spatial index $i$ in $14 \times 14$ grids.

Lu et al. [15] propose an element-wise multiplication of a question vector and a visual feature vector after appropriate embeddings for a joint model. This makes a strong baseline outperforming some of the recent works [19, 2]. We firstly take this approach as a candidate for the joint residual function, since it is simple yet successful for visual question-answering. In this context, we take the global visual feature approach for the element-wise multiplication, instead of the multiple (spatial) visual features approach for the explicit attention mechanism of SAN. (We present a visualization technique exploiting the element-wise multiplication in Section 5.2.)

Based on these observations, we follow the shortcut mapping and the stacking architecture of SAN [29]; however, the element-wise multiplication is used for the joint residual function $\mathcal{F}$. These updates effectively learn the joint representation of given vision and language information addressing the *bottleneck* issue of the attention networks of SAN.

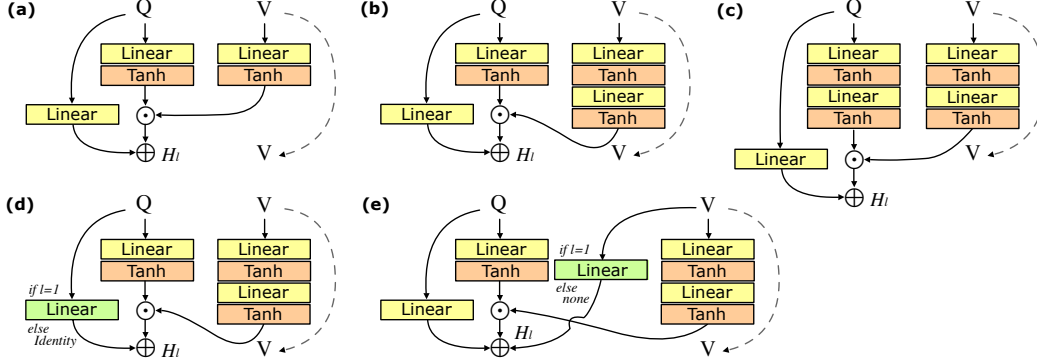

Figure 3: Alternative models are explored to justify our proposed model. The base model (a) has a shortcut for a question vector as SAN does [29], and the joint residual function takes the form of the *Deep Q+I* model's joint function [15]. (b) extra embedding for visual modality. (c) extra embeddings for both modalities. (d) identity mappings for shortcuts. In the first learning block, use a linear mapping for matching a dimension with the joint dimension. (e) two shortcuts for both modalities. For simplicity, the linear mapping of visual shortcut only appears in the first learning block. Notice that (d) and (e) are compared to (b) after the model selection of (b) among (a)-(c) on *test-dev* results. Eventually, we chose (b) as the best performance and relative simplicity.

## 3.2 Multimodal Residual Networks

MRN consists of multiple learning blocks, which are stacked for deep residual learning. Denoting an optimal mapping by $\mathcal{H}(\mathbf{q}, \mathbf{v})$, we approximate it using

$$H_1(\mathbf{q}, \mathbf{v}) = W_{\mathbf{q}'}^{(1)}\mathbf{q} + \mathcal{F}^{(1)}(\mathbf{q}, \mathbf{v}). \tag{4}$$

The first (linear) approximation term is $W_{\mathbf{q}'}^{(1)}\mathbf{q}$ and the first joint residual function is given by $\mathcal{F}^{(1)}(\mathbf{q}, \mathbf{v})$. The linear mapping $W_{\mathbf{q}'}$ is used for matching a feature dimension. We define the joint residual function as

$$\mathcal{F}^{(k)}(\mathbf{q}, \mathbf{v}) = \sigma(W_{\mathbf{q}}^{(k)}\mathbf{q}) \odot \sigma(W_2^{(k)}\sigma(W_1^{(k)}\mathbf{v})) \tag{5}$$

where $\sigma$ is $\tanh$, and $\odot$ is element-wise multiplication. The question vector and the visual feature vector directly contribute to the joint representation. We justify this choice in Sections 4 and 5.

For a deeper residual learning, we replace $\mathbf{q}$ with $H_1(\mathbf{q}, \mathbf{v})$ in the next layer. In more general terms, Equations 4 and 5 can be rewritten as

$$H_L(\mathbf{q}, \mathbf{v}) = W_{\mathbf{q}'}\mathbf{q} + \sum_{l=1}^{L} W_{\mathcal{F}^{(l)}}\mathcal{F}^{(l)}(H_{l-1}, \mathbf{v}) \tag{6}$$

where $L$ is the number of learning blocks, $H_0 = \mathbf{q}$, $W_{\mathbf{q}'} = \Pi_{l=1}^{L}W_{\mathbf{q}'}^{(l)}$, and $W_{\mathcal{F}^{(l)}} = \Pi_{m=l+1}^{L}W_{\mathbf{q}'}^{(m)}$. The cascading in Equation 6 can intuitively be represented as shown in Figure 2. Notice that the shortcuts for a visual part are identity mappings to transfer the input visual feature vector to each layer (dashed line). At the end of each block, we denote $H_l$ as the output of the $l$-th learning block, and $\oplus$ is element-wise addition.

## 4 Experiments

### 4.1 Visual QA Dataset

We choose the Visual QA (VQA) dataset [1] for the evaluation of our models. Other datasets may not be ideal, since they have limited number of examples to train and test [16], or have synthesized questions from the image captions [14, 20].

Table 1: The results of alternative models (a)-(e) on the *test-dev*.

|     | Open-Ended | | | |
|-----|-------|-------|-------|-------|
|     | All   | Y/N   | Num.  | Other |
| (a) | 60.17 | 81.83 | 38.32 | 46.61 |
| (b) | **60.53** | **82.53** | **38.34** | **46.78** |
| (c) | 60.19 | 81.91 | 37.87 | 46.70 |
| (d) | 59.69 | 81.67 | 37.23 | 46.00 |
| (e) | 60.20 | 81.98 | 38.25 | 46.57 |

Table 2: The effect of the visual features and # of target answers on the *test-dev* results. *Vgg* for *VGG-19*, and *Res* for *ResNet-152* features described in Section 4.

|          | Open-Ended | | | |
|----------|-------|-------|-------|-------|
|          | All   | Y/N   | Num.  | Other |
| *Vgg*, 1k | 60.53 | **82.53** | 38.34 | 46.78 |
| *Vgg*, 2k | 60.77 | 82.10 | **39.11** | 47.46 |
| *Vgg*, 3k | 60.68 | 82.40 | 38.69 | 47.10 |
| *Res*, 1k | 61.45 | 82.36 | 38.40 | 48.81 |
| *Res*, 2k | **61.68** | 82.28 | 38.82 | **49.25** |
| *Res*, 3k | 61.47 | 82.28 | 39.09 | 48.76 |

The questions and answers of the VQA dataset are collected via Amazon Mechanical Turk from human subjects, who satisfy the experimental requirement. The dataset includes 614,163 questions and 7,984,119 answers, since ten answers are gathered for each question from unique human subjects. Therefore, Agrawal et al. [1] proposed a new accuracy metric as follows:

$$\min\left(\frac{\text{\# of humans that provided that answer}}{3}, 1\right). \tag{7}$$

The questions are answered in two ways: *Open-Ended* and *Multiple-Choice*. Unlike *Open-Ended*, *Multiple-Choice* allows additional information of eighteen candidate answers for each question. There are three types of answers: yes/no (*Y/N*), numbers (*Num.*) and others (*Other*). Table 3 shows that *Other* type has the most benefit from *Multiple-Choice*.

The images come from the MS-COCO dataset, 123,287 of them for training and validation, and 81,434 for test. The images are carefully collected to contain multiple objects and natural situations, which is also valid for visual question-answering tasks.

## 4.2 Implementation

Torch framework and *rnn* package [13] are used to build our models. For efficient computation of variable-length questions, *TrimZero* is used to trim out zero vectors [11]. TrimZero eliminates zero computations at every time-step in mini-batch learning. Its efficiency is affected by a batch size, RNN model size, and the number of zeros in inputs. We found out that TrimZero was suitable for VQA tasks. Approximately, 37.5% of training time is reduced in our experiments using this technique.

**Preprocessing**   We follow the same preprocessing procedure of DeeperLSTM+NormalizedCNN [15] (*Deep Q+I*) by default. The number of answers is 1k, 2k, or 3k using the most frequent answers, which covers 86.52%, 90.45% and 92.42% of questions, respectively. The questions are tokenized using Python Natural Language Toolkit (nltk) [3]. Subsequently, the vocabulary sizes are 14,770, 15,031 and 15,169, respectively.

**Pretrained Models**   A question vector $\mathbf{q} \in \mathbb{R}^{2,400}$ is the last output vector of GRU [4], initialized with the parameters of Skip-Thought Vectors [12]. Based on the study of Noh et al. [19], this method shows effectiveness of question embedding in visual question-answering tasks. A visual feature vector $\mathbf{v}$ is an output of the first fully-connected layer of *VGG-19* networks [23], whose dimension is 4,096. Alternatively, *ResNet-152* [6] is used, whose dimension is of 2,048. The error is back-propagated to the input question for fine-tuning, yet, not for the visual part $\mathbf{v}$ due to the heavy computational cost of training.

**Postprocessing**   Image captioning model [10] is used to improve the accuracy of *Other* type. Let the intermediate representation $\mathbf{v} \in \mathbb{R}^{|\Omega|}$ which is right before applying softmax. $|\Omega|$ is the vocabulary size of answers, and $\mathbf{v}_i$ is corresponding to answer $\mathbf{a}_i$. If $\mathbf{a}_i$ is not a number or *yes* or *no*, and appeared at least once in the generated caption, then update $\mathbf{v}_i \leftarrow \mathbf{v}_i + 1$. Notice that the pretrained image captioning model is not part of training. This simple procedure improves around 0.1% of the *test-dev*

Table 3: The VQA *test-standard* results. The precision of some accuracies [29, 2] are one less than others, so, zero-filled to match others.

| | Open-Ended | | | | Multiple-Choice | | | |
|---|---|---|---|---|---|---|---|---|
| | All | Y/N | Num. | Other | All | Y/N | Num. | Other |
| DPPnet [19] | 57.36 | 80.28 | 36.92 | 42.24 | 62.69 | 80.35 | 38.79 | 52.79 |
| D-NMN [2] | 58.00 | - | - | - | - | - | - | - |
| Deep Q+I [15] | 58.16 | 80.56 | 36.53 | 43.73 | 63.09 | 80.59 | 37.70 | 53.64 |
| SAN [29] | 58.90 | - | - | - | - | - | - | - |
| ACK [27] | 59.44 | 81.07 | 37.12 | 45.83 | - | - | - | - |
| FDA [8] | 59.54 | 81.34 | 35.67 | 46.10 | 64.18 | 81.25 | 38.30 | 55.20 |
| DMN+ [28] | 60.36 | 80.43 | 36.82 | 48.33 | - | - | - | - |
| MRN | **61.84** | **82.39** | **38.23** | **49.41** | **66.33** | **82.41** | **39.57** | **58.40** |
| Human [1] | 83.30 | 95.77 | 83.39 | 72.67 | - | - | - | - |

overall accuracy (0.3% for *Other* type). We attribute this improvement to "tie break" in *Other* type. For the *Multiple-Choice* task, we mask the output of softmax layer with the given candidate answers.

**Hyperparameters** By default, we follow *Deep Q+I*. The common embedding size of the joint representation is 1,200. The learnable parameters are initialized using a uniform distribution from $-0.08$ to $0.08$ except for the pretrained models. The batch size is 200, and the number of iterations is fixed to 250k. The RMSProp [26] is used for optimization, and dropouts [7, 5] are used for regularization. The hyperparameters are fixed using *test-dev* results. We compare our method to state-of-the-arts using *test-standard* results.

## 4.3 Exploring Alternative Models

Figure 3 shows alternative models we explored, based on the observations in Section 3. We carefully select alternative models (a)-(c) for the importance of embeddings in multimodal learning [18, 24], (d) for the effectiveness of identity mapping as reported by [6], and (e) for the confirmation of using question-only shortcuts in the multiple blocks as in [29]. For comparison, all models have three-block layers (selected after a pilot test), using *VGG-19* features and 1k answers, then, the number of learning blocks is explored to confirm the pilot test. The effect of the pretrained visual feature models and the number of answers are also explored. All validation is performed on the *test-dev* split.

## 5 Results

### 5.1 Quantitative Analysis

The VQA Challenge, which released the VQA dataset, provides evaluation servers for *test-dev* and *test-standard* test splits. For the *test-dev*, the evaluation server permits unlimited submissions for validation, while the *test-standard* permits limited submissions for the competition. We report accuracies in percentage.

**Alternative Models** The *test-dev* results of the alternative models for the *Open-Ended* task are shown in Table 1. (a) shows a significant improvement over SAN. However, (b) is marginally better than (a). As compared to (b), (c) deteriorates the performance. An extra embedding for a question vector may easily cause overfitting leading to the overall degradation. And, the identity shortcuts in (d) cause the degradation problem, too. Extra parameters of the linear mappings may effectively support to do the task.

(e) shows a reasonable performance, however, the extra shortcut is not essential. The empirical results seem to support this idea. Since the question-only model (50.39%) achieves a competitive result to the joint model (57.75%), while the image-only model gets a poor accuracy (28.13%) (see Table 2 in [1]). Eventually, we chose model (b) as the best performance and relative simplicity.

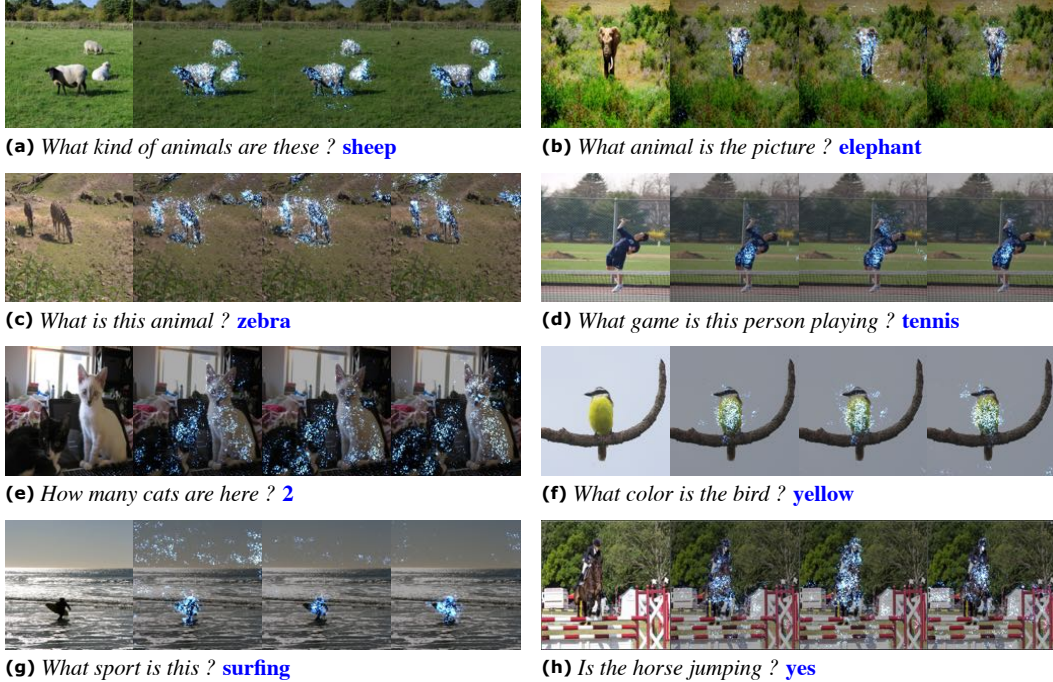

**(a)** *What kind of animals are these ?* **sheep**

**(b)** *What animal is the picture ?* **elephant**

**(c)** *What is this animal ?* **zebra**

**(d)** *What game is this person playing ?* **tennis**

**(e)** *How many cats are here ?* **2**

**(f)** *What color is the bird ?* **yellow**

**(g)** *What sport is this ?* **surfing**

**(h)** *Is the horse jumping ?* **yes**

Figure 4: Examples for visualization of the three-block layered MRN. The original images are shown in the first of each group. The next three images show the input gradients of the attention effect for each learning block as described in Section 5.2. The gradients of color channels for each pixel are summed up after taking absolute values of these gradients. Then, these summed absolute values which are greater than the summation of the mean and the standard deviation of these values are visualized as the attention effect (bright color) on the images. The answers (blue) are predicted by MRN.

The effects of other various options, Skip-Thought Vectors [12] for parameter initialization, Bayesian Dropout [5] for regularization, image captioning model [10] for postprocessing, and the usage of shortcut connections, are explored in Appendix A.1.

**Number of Learning Blocks**   To confirm the effectiveness of the number of learning blocks selected via a pilot test ($L = 3$), we explore this on the chosen model (b), again. As the depth increases, the overall accuracies are 58.85% ($L = 1$), 59.44% ($L = 2$), **60.53%** ($L = 3$) and 60.42% ($L = 4$).

**Visual Features**   The ResNet-152 visual features are significantly better than *VGG-19* features for *Other* type in Table 2, even if the dimension of the ResNet features (2,048) is a half of *VGG* features' (4,096). The *ResNet* visual features are also used in the previous work [8]; however, our model achieves a remarkably better performance with a large margin (see Table 3).

**Number of Target Answers**   The number of target answers slightly affects the overall accuracies with the trade-off among answer types. So, the decision on the number of target answers is difficult to be made. We chose *Res*, 2k in Table 2 based on the overall accuracy (for *Multiple-Choice* task, see Appendix A.1).

**Comparisons with State-of-the-arts**   Our chosen model significantly outperforms other state-of-the-art methods for both *Open-Ended* and *Multiple-Choice* tasks in Table 3. However, the performance of *Number* and *Other* types are still not satisfactory compared to *Human* performance, though the advances in the recent works were mainly for *Other*-type answers. This fact motivates to study on a counting mechanism in future work. The model comparison is performed on the *test-standard* results.

## 5.2 Qualitative Analysis

In Equation 5, the left term $\sigma(W_\mathbf{q}\mathbf{q})$ can be seen as a masking (attention) vector to select a part of visual information. We assume that the difference between the right term $\mathcal{V} := \sigma(W_2\sigma(W_1\mathbf{v}))$ and the masked vector $\mathcal{F}(\mathbf{q}, \mathbf{v})$ indicates an attention effect caused by the masking vector. Then, the attention effect $\mathcal{L}_{att} = \frac{1}{2}\|\mathcal{V} - \mathcal{F}\|^2$ is visualized on the image by calculating the gradient of $\mathcal{L}_{att}$ with respect to a given image $\mathcal{I}$, while treating $\mathcal{F}$ as a constant.

$$\frac{\partial \mathcal{L}_{att}}{\partial \mathcal{I}} = \frac{\partial \mathcal{V}}{\partial \mathcal{I}}(\mathcal{V} - \mathcal{F}) \tag{8}$$

This technique can be applied to each learning block in a similar way.

Since we use the preprocessed visual features, the pretrained CNN is augmented only for this visualization. Note that model (b) in Table 1 is used for this visualization, and the pretrained *VGG-19* is used for preprocessing and augmentation. The model is trained using the training set of the VQA dataset, and visualized using the validation set. Examples are shown in Figure 4 (more examples in Appendix A.2-4).

Unlike the other works [29, 28] that use explicit attention parameters, MRN does not use any explicit attentional mechanism. However, we observe the interpretability of element-wise multiplication as an information masking, which yields a novel method for visualizing the attention effect from this operation. Since MRN does not depend on a few attention parameters (*e.g.* $14 \times 14$), our visualization method shows a higher resolution than others [29, 28]. Based on this, we argue that MRN is an implicit attention model without explicit attention mechanism.

## 6 Conclusions

The idea of deep residual learning is applied to visual question-answering tasks. Based on the two observations of the previous works, various alternative models are suggested and validated to propose the three-block layered MRN. Our model achieves the state-of-the-art results on the VQA dataset for both *Open-Ended* and *Multiple-Choice* tasks. Moreover, we have introduced a novel method to visualize the spatial attention from the collapsed visual features using back-propagation.

We believe our visualization method brings implicit attention mechanism to research of attentional models. Using back-propagation of attention effect, extensive research in object detection, segmentation and tracking are worth further investigations.

### Acknowledgments

The authors would like to thank Patrick Emaase for helpful comments and editing. This work was supported by Naver Corp. and partly by the Korea government (IITP-R0126-16-1072-SW.StarLab, KEIT-10044009-HRI.MESSI, KEIT-10060086-RISF, ADD-UD130070ID-BMRR).

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
