[Supplementary Material]

# A Appendix

## A.1 VQA *test-dev* Results

Table 1: The effects of various options for VQA *test-dev*. Here, the model of Figure 3a is used, since these experiments are preliminarily conducted. *VGG-19* features and 1k target answers are used. *s* stands for the usage of Skip-Thought Vectors [6] to initialize the question embedding model of GRU, *b* stands for the usage of Bayesian Dropout [3], and *c* stands for the usage of postprocessing using image captioning model [5].

|  | Open-Ended | | | | Multiple-Choice | | | |
|---|---|---|---|---|---|---|---|---|
|  | All | Y/N | Num. | Other | All | Y/N | Num. | Other |
| *baseline* | 58.97 | 81.11 | 37.63 | 44.90 | 63.53 | 81.13 | 38.91 | 54.06 |
| *s* | 59.38 | 80.65 | **38.30** | 45.98 | 63.71 | 80.68 | **39.73** | 54.65 |
| *s,b* | 59.74 | **81.75** | 38.13 | 45.84 | 64.15 | **81.77** | 39.54 | 54.67 |
| *s,b,c* | **59.91** | **81.75** | 38.13 | **46.19** | **64.18** | **81.77** | 39.51 | **54.72** |

Table 2: The results for VQA *test-dev*. The precision of some accuracies [11, 2, 10] are one less than others, so, zero-filled to match others.

|  | Open-Ended | | | | Multiple-Choice | | | |
|---|---|---|---|---|---|---|---|---|
|  | All | Y/N | Num. | Other | All | Y/N | Num. | Other |
| Question [1] | 48.09 | 75.66 | 36.70 | 27.14 | 53.68 | 75.71 | 37.05 | 38.64 |
| Image [1] | 28.13 | 64.01 | 00.42 | 03.77 | 30.53 | 69.87 | 00.45 | 03.76 |
| Q+I [1] | 52.64 | 75.55 | 33.67 | 37.37 | 58.97 | 75.59 | 34.35 | 50.33 |
| LSTM Q [1] | 48.76 | 78.20 | 35.68 | 26.59 | 54.75 | 78.22 | 36.82 | 38.78 |
| LSTM Q+I [1] | 53.74 | 78.94 | 35.24 | 36.42 | 57.17 | 78.95 | 35.80 | 43.41 |
| Deep Q+I [7] | 58.02 | 80.87 | 36.46 | 43.40 | 62.86 | 80.88 | 37.78 | 53.14 |
| DPPnet [8] | 57.22 | 80.71 | 37.24 | 41.69 | 62.48 | 80.79 | 38.94 | 52.16 |
| D-NMN [2] | 57.90 | 80.50 | 37.40 | 43.10 | - | - | - | - |
| SAN [11] | 58.70 | 79.30 | 36.60 | 46.10 | - | - | - | - |
| ACK [9] | 59.17 | 81.01 | 38.42 | 45.23 | - | - | - | - |
| FDA [4] | 59.24 | 81.14 | 36.16 | 45.77 | 64.01 | 81.50 | 39.00 | 54.72 |
| DMN+ [10] | 60.30 | 80.50 | 36.80 | 48.30 | - | - | - | - |
| *Vgg*, 1k | 60.53 | **82.53** | 38.34 | 46.78 | 64.79 | **82.55** | 39.93 | 55.23 |
| *Vgg*, 2k | 60.77 | 82.10 | **39.11** | 47.46 | 65.27 | 82.12 | **40.84** | 56.39 |
| *Vgg*, 3k | 60.68 | 82.40 | 38.69 | 47.10 | 65.09 | 82.42 | 40.13 | 55.93 |
| *Res*, 1k | 61.45 | 82.36 | 38.40 | 48.81 | 65.62 | 82.39 | 39.65 | 57.15 |
| *Res*, 2k | **61.68** | 82.28 | 38.82 | **49.25** | 66.15 | 82.30 | 40.45 | 58.16 |
| *Res*, 3k | 61.47 | 82.28 | 39.09 | 48.76 | **66.33** | 82.41 | 39.57 | **58.40** |

Table 3: The effects of shortcut connections of MRN for VQA *test-dev*. *ResNet-152* features and 2k target answers are used. *MN* stands for Multimodal Networks without residual learning, which does not have any shortcut connections. *Dim.* stands for common embedding vector's dimension. The number of parameters for word embedding (9.3M) and question embedding (21.8M) is subtracted from the total number of parameters in this table.

|  | L | Dim. | #params | Open-Ended | | | |
|---|---|---|---|---|---|---|---|
|  |  |  |  | All | Y/N | Num. | Other |
| MN | 1 | 4604 | 33.9M | 60.33 | **82.50** | 36.04 | 46.89 |
| MN | 2 | 2350 | 33.9M | 60.90 | 81.96 | 37.16 | 48.28 |
| MN | 3 | 1559 | 33.9M | 59.87 | 80.55 | 37.53 | 47.25 |
| MRN | 1 | 3355 | 33.9M | 60.09 | 81.78 | 37.09 | 46.78 |
| MRN | 2 | 1766 | 33.9M | 61.05 | 81.81 | 38.43 | 48.43 |
| MRN | 3 | 1200 | 33.9M | **61.68** | 82.28 | 38.82 | **49.25** |
| MRN | 4 | 851 | 33.9M | 61.02 | 82.06 | **39.02** | 48.04 |

## A.2 More Examples

**(a)** *Does the man have good posture ?* **no**

**(b)** *Did he fall down ?* **yes**

**(c)** *Are there two cats in the picture ?* **no**

**(d)** *What color are the bears ?* **brown**

**(e)** *What are many of the people carrying ?* **umbrellas**

**(f)** *What color is the dog ?* **black**

**(g)** *Are these animals tall ?* **yes**

**(h)** *What animal is that ?* **sheep**

**(i)** *Are all the cows the same color ?* **no**

**(j)** *What is the reflection of in the mirror ?* **dog**

**(k)** *What are the giraffe in the foreground doing ?* **eating**

**(l)** *What animal is standing in the water other than birds ?* **bear**

Figure 1: More examples of Figure 4 in Section 5.2.

## A.3 Comparative Analysis

**(a1)** *What is the animal on the left ?* **giraffe**

**(a2)** *Can you see trees ?* **yes**

**(b1)** *What is the lady riding ?* **motorcycle**

**(b2)** *Is she riding the motorcycle on the street ?* **no**

Figure 2: Comparative examples on the same image. (a1) and (a2) depict a giraffe (left) and a man pointing at the giraffe. MRN consistently highlights on the giraffe in (a1). However, the other question *"Can you see trees?"* makes MRN less attentive to the giraffe, while a tree in the right of background is more focused in (a2). Similarily, the attention effect of (b2) is widely dispersed on background than (b1) in the middle of sequences, may be to recognize the site. However, the subtlety in comparative study is insufficient to objectively assess the results.

## A.4 Failure Examples

**(a)** *What animals are these ?* **bears** **ducks**

**(b)** *What are these animals ?* **cows** **goats**

**(c)** *What animals are visible ?* **sheep** **horses**

**(d)** *How many animals are depicted ?* **2 1**

**(e)** *What flavor donut is this ?* **chocolate** **strawberry**

**(f)** *What is the man doing ?* **playing tennis** **frisbee**

**(g)** *What color are the giraffes eyelashes ?* **brown** **black**

**(h)** *What food is the bear trying to eat ?* **banana** **papaya**

**(i)** *What kind of animal is used to herd these animals ?* **sheep** **dog**

**(j)** *What species of tree are in the background ?* **pine** **palm**

**(k)** *Are there any birds on the photo ?* **no** **yes**

**(l)** *Why is the hydrant smiling ?* **happy** **someone drew on it**

Figure 3: Failure Examples. Each question is followed by model prediction (blue) and answer (red). As mentioned in Section 5, MRN shows the weakness of counting in (d) and (k). Sometimes, the model finds objects regardless of the given question. In (j), even if a word *cat* does not appear in the question, the cat in the image is surely attended. (i) shows the limitation of attentional mechanism, which needs an inference using world knowledge.