[Reviews · NeurIPS 2016]

Reviewer 1

Summary

This paper combines the residual like network with multimodal fusion of language and visual features for Visual QA task. Impressive results on standard benchmarks show progress. While the novelty is limited, accepting the paper will help others build on the state-of-the-art results.

Qualitative Assessment

This paper proposes to combine residual like architecture with multimodal fusion for visual question answering. A particular novelty is the usage of multiplicative interactions to combine visual features with word embeddings. The results are impressive, betting the state-of-the-art by a margin. However, this paper used many pretrained models and embeddings, so it would make the paper better if all these effects are better analyzed. It is better not not refer to equations in other papers. For example in section 3.1, it is better if the equations are reproduced in this paper. Question: In section 5.2, what happens if you use sigmoid(W_q*q) as the attentional mask for visualization?

Confidence in this Review

2-Confident (read it all; understood it all reasonably well)


Reviewer 2

Summary

The authors proposed a new network architecture to extend the deep residual learning to multi-modal inputs, applied to a visual question-answering task. Unlike some previous work where explicit attention network mechanism was utilized, the proposed method follows a previous work using element-wise multiplication between question and visual feature vectors for the joint residual function representation, seen as an implicit attention mechanism. The authors experimented with multiple variants of the architecture and settled on a relatively simple learning block in a 3-block layered network. The combination of deep residual learning and implicit attention were shown to be effective on the Visual QA data set and out-performed other state-of-the-art results. To visualize the attention effect of multi-modal residual learning, the authors proposed a technique to display the difference between visual input and the joint residual mapping with back-propagation for each learning block. Examples from visual QA task intuitively demonstrated the implicit attention effect.

Qualitative Assessment

The authors successfully built upon two effective ideas, the deep residual learning and element-wise multiplication for implicit attention, and created a solution for general multi-modal tasks. Experiments were carefully run to select an optimal architecture and hyper-parameters for the targeted Visual QA task. The results appeared to be superb, compared to previous studies with various deep learning techniques. It would be helpful if the authors can present additional comparison with existing techniques in terms of model parameter size, as well as amount of data required for learning. It would also be interesting to separately assess the value of residual learning and implicit attention on the Visual QA task, to help understand which aspect is the most critical. The proposed visualization method was intuitively appealing and the examples demonstrated its effectiveness in explaining the implicit attention mechanism. It would be helpful if the authors can include additional explanation on the differences between the three images at each block layer, and perhaps provide an intuitive explanation why three layer appears to be optimal for this task.

Confidence in this Review

2-Confident (read it all; understood it all reasonably well)


Reviewer 3

Summary

This paper proposed a Mutimodal method based residual network for visual question-answering. As a multimodal method, this work is able to learn the joint representation from visual and language information. The state-of-art results on the data set demonstrate the effectiveness of this proposed work. More than that, the introduced method to visualize attention effects of the joint representation is interesting. Overall, this paper is well-written and easy to read. Proposed idea has been validated using relevant experiments. There are some latest papers on VQA topic, if possible, please make some comparisons with the latest published papers. For example, ICML 2016 "Dynamic Memory Networks for Visual and Textual Question Answering"

Qualitative Assessment

This paper proposed a Mutimodal method based residual network for visual question-answering. As a multimodal method, this work is able to learn the joint representation from visual and language information. The state-of-art results on the data set demonstrate the effectiveness of this proposed work. More than that, the introduced method to visualize attention effects of the joint representation is interesting and seems effective. There are some latest papers working on Visual Question-Asnwering, for example, ICML 2016 "Dynamic Memory Networks for Visual and Textual Question Answering". It is suggested to make comparisons with these latest methods.

Confidence in this Review

1-Less confident (might not have understood significant parts)


Reviewer 4

Summary

The paper proposes to use element-wise multiplication for the joint residual mapping for VQA.

Qualitative Assessment

The quality of the paper is not good enough for publication for the reasons below: - performance is not state-of-the-art. - idea is not novel. Similar framework have been explored in [1]. [1] Fukui, A., Park, D. H., Yang, D., Rohrbach, A., Darrell, T., & Rohrbach, M. (2016). Multimodal Compact Bilinear Pooling for Visual Question Answering and Visual Grounding. Retrieved from http://arxiv.org/abs/1606.01847

Confidence in this Review

2-Confident (read it all; understood it all reasonably well)


Reviewer 5

Summary

The paper presents a multimodal learning scheme in lines of residual learning for addressing visual question answering problem. It achieves state of the art results on VQA. The most interesting part is the attention representation using back propagation without having attention parameters.

Qualitative Assessment

If I understand correctly, the most interesting part is visualizing VQA attention without having attention parameters. Some background references on VQA will make the manuscript better for a non-VQA reader. With VQA becoming popular, information in section 4.1 seems redundant and taking lot of space. Some explanation on TrimZero in lines 136-137 may be helpful. In section 4.2- postprocessing, the update v= v+1 (line 154) is not clear. The results are impressive.

Confidence in this Review

2-Confident (read it all; understood it all reasonably well)


Reviewer 6

Summary

This paper proposed Multimodal Residual Networks (MRN) for the multimodal residual learning of visual question-answering. The framework utilized CNN models for extracting visual features and RNN models for language information processing. Then a MRN model can effectively learns the joint representation from visual and language information. They achieve the state-of-the-art results on the Visual QA dataset for both Open-Ended and Multiple-Choice tasks. They also introduce a novel method to visualize the attention effect of the joint representations.

Qualitative Assessment

The MRN is effective for Visual QA dataset or visual question-answering task. However, contributions are little simple. As far as I know, in order to learn the joint representation, the using of DNN is very common. This paper did not analyze which component of the framework contributes the most. Is CNN or RNN models that they have used or the MRN? If there is any proof or discussion, this will be an impressive paper.

Confidence in this Review

2-Confident (read it all; understood it all reasonably well)